# Budget Impact Analysis of Adopting a One-Step Nucleic Acid Amplification Testing (NAAT) Alone Diagnostic Pathway for *Clostridioides difficile* in Japan Compared to a Two-Step Algorithm with Glutamate Dehydrogenase/Toxin Followed by NAAT

**DOI:** 10.3390/diagnostics13081463

**Published:** 2023-04-18

**Authors:** Vanessa W. Lim, Takeshi Tomaru, Brandon Chua, Yan Ma, Katsunori Yanagihara

**Affiliations:** 1Health Economics and Outcomes Research, Becton Dickinson Holdings Pte. Ltd., 2 International Business Park Road, Singapore 609930, Singapore; 2Health Economics and Outcomes Research, Nippon Becton Dickinson Company, Ltd., Akasaka Garden City 15-1, Akasaka 4-Chome, Minato-ku, Tokyo 107-0052, Japan; 3Saw Swee Hock School of Public Health, National University of Singapore, 12 Science Drive 2, #10-02, Singapore 117549, Singapore; 4Department of Laboratory Medicine, Nagasaki University Graduate School of Biomedical Sciences, Nagasaki University Hospital, Sakamoto 1-12-4, Nagasaki City 852-8523, Japan

**Keywords:** budget impact analysis, *Clostridioides difficile* infection, Japan, healthcare associated infections, diagnostic pathway, nucleic acid amplification testing, NAAT

## Abstract

*Clostridioides difficile* infection (CDI) is a major healthcare-associated infection that leads to a significant health economic burden in Japan. Using a decision tree model, we evaluated the budget impact of adopting a one-step nucleic acid amplification test (NAAT) alone pathway compared to a two-step diagnostic algorithm with glutamate dehydrogenase (GDH) and toxin antigen, followed by NAAT. The analysis was conducted from the government payer’s perspective for 100,000 symptomatic, hospitalized adults requiring a CDI diagnostic test. One-way sensitivity analysis was conducted for all data inputs. The NAAT alone strategy costed JPY 225,886,360 (USD 2,424,714) more, but was more effective, resulting in 1749 more patients accurately diagnosed and 91 fewer deaths compared to the two-step algorithm. Additionally, the NAAT alone pathway costed JPY 26,146 (USD 281) less per true positive CDI diagnosed. The total budget impact, and cost per CDI diagnosed was most sensitive to GDH sensitivity in one-way sensitivity analysis, where a lower GDH sensitivity resulted in greater cost savings with the NAAT alone pathway. Findings from this budget impact analysis can guide the adoption of a NAAT alone pathway for CDI diagnosis in Japan.

## 1. Introduction

Globally, *Clostridioides difficile* infection (CDI) is one of the most common hospital-acquired infections [1] that represents a significant economic burden on the healthcare system [2]. In Japan, the prevalence of CDI ranges between 0.3 to 5.5 per 1000 patients [3]. Each CDI increases the inpatient expenditure of hospitalized patients by more than 320,000 Japanese yen (JPY) [4]. In hospital settings, *Clostridioides difficile* is most commonly transmitted via direct contact with CDI patients and the contaminated hospital environment, followed by contact with asymptomatic patients colonized with *Clostridioides difficile* [5]. Hence, infection control measures are key to prevent transmission of CDI [6,7]. This includes the accurate and timely diagnosis of CDI to initiate appropriate treatment, improve patient outcomes, and reduce the economic burden to the hospital and payers [8,9]. Besides that, judicious use of antimicrobial therapy is key, because the use of antimicrobials promotes the emergence of CDI [10].

Since 2017, the Infectious Diseases Society of America (IDSA) and Society for Healthcare Epidemiology of America (SHEA) has adopted the use of nucleic acid amplification tests (NAATs) for CDI diagnosis [11]. Subsequently in 2019, NAAT was also recommended by the American Society for Microbiology for CDI diagnosis [12]. This includes using NAAT alone, or NAAT in combination with either glutamate dehydrogenase (GDH) alone or GDH plus toxin enzyme immunoassay [11,12]. However, multiple pieces of evidence have shown that first-line GDH-based algorithms are less sensitive compared to NAAT alone algorithms. One prospective analysis of over 1000 stool specimens from patients with clinically suspected CDI showed that the sensitivity of NAAT alone was 99.1% compared to 83.8% for GDH alone [13]. Furthermore, rapid diagnosis of CDI with NAAT alone was more timely and accurate, and correlated better with clinical diagnosis [13]. In another study among pediatric patients, the sensitivity of a NAAT alone algorithm was 94% versus 85% for the GDH/toxin algorithm, suggesting that GDH-based algorithms can result in more false negative or missed CDI diagnoses [14]. In addition to diagnostic inaccuracy, evidence suggests that CDI has been underdiagnosed in Japan by up to 24% based on current practice [15,16], where GDH/toxin is the first-line test for CDI. This is likely due to a lack of clinical suspicion and limitations in microbiological testing for CDI [16].

A missed CDI diagnosis can have a significant economic impact compared to those who were accurately diagnosed with the initial test. In a study conducted in United States, patients with a missed CDI diagnosis had an additional 7 days of hospital stay on average, depending on the repeat testing frequency in the hospital [8]. In contrast, the use of NAAT alone for CDI diagnosis led to fewer days of empiric antibiotic therapy for patients without CDI [17]. This supports antimicrobial stewardship practices, which aim to reduce unnecessary antibiotic use [18]. Japan has a long average hospital length of stay of 16.4 days compared to the rest of the world [19], which itself is a known risk factor of CDI [20]. In addition, Japan has a long CDI-attributable length of hospital stay of 11.96 days [4], which contributes towards a significant economic burden to the health system [21], possibly even more than what we see in the US.

Since January 2023, the Japan CDI clinical practice guideline has been revised by the Japanese Association of Chemotherapy to include a one-step NAAT alone pathway as a diagnostic option for CDI, along with fidaxomicin as a treatment option [22]. In the current study, we aimed to evaluate the budgetary impact of a one-step NAAT alone pathway in Japan, by comparing it with a two-step CDI diagnostic algorithm initially with GDH/toxin, followed by NAAT.

## 2. Materials and Methods

### 2.1. Model Overview

A decision tree was constructed to compare two CDI diagnostic approaches (Microsoft Excel^®^ 2016) from the Japan government payer’s perspective: (1) one-step pathway with NAAT alone; (2) two-step algorithm with GDH/toxin followed by NAAT (stool samples were first tested with GDH and toxin; where GDH positive, toxin-negative specimens were then tested via NAAT). Figure 1 shows the CDI diagnostic pathway according to the revised guidelines in Japan [22].

We simulated a total of 100,000 symptomatic, hospitalized adult patients suspected with CDI requiring a CDI diagnostic test. Based on Japanese clinical practice, we modeled that patients with at least three diarrheal bowel movements (Bristol stool grade ≥5) in the preceding 24 h would be tested and treated [22]. Up to two CDI recurrences were modelled for patients after the initial CDI, similar to a previous study evaluating the cost-effectiveness of CDI diagnostics in the United States [23]. Since CDI recurrences usually occur ≤60 days after the cure date of the initial CDI episode [24], the time horizon of this model was one year. In the NAAT alone pathway, patients tested positive with NAAT (NAAT+) would be treated immediately. In the two-step algorithm, patients would be treated when both GDH and toxin was positive (GDH+/toxin+), or when GDH alone was positive (GDH+/toxin−) followed by NAAT+.

The definitions of each test outcome are as follows: (1) True positive refers to a positive test for CDI in the presence of CDI; (2) False positive refers to a positive test in the absence of CDI; (3) True negative refers to a negative CDI test in the absence of CDI; (4) False negative refers to a negative test in the presence of CDI. Hence, a true positive CDI was deemed as an accurately diagnosed CDI in this study, while a false negative CDI was deemed as a missed CDI diagnosis. False negative CDI patients were modeled with the same pathway as true positive CDI patients. After CDI treatment, patients could be (1) cured with the possibility of recurrence; (2) unresponsive to treatment (treatment failure); or (3) dead due to CDI. Patients with treatment failure were given intravenous metronidazole and vancomycin as the last line of drug treatment based on recommendations provided by the Ministry of Health, Labour and Welfare [25]. Patients who failed the drug treatment were subjected to subtotal colectomy. The model schematic detailing the full diagnostic pathway is shown in Figure 2. The total cost and clinical outcomes, including number of true positive patients diagnosed and number of deaths, were calculated for each diagnostic pathway. Besides determining the total cost difference between NAAT alone and the two-step algorithm, we also compared the cost per true positive CDI diagnosed in each pathway, by dividing the total cost by the total true positive CDI diagnosed.

### 2.2. Clinical and Diagnostic Data Inputs

The clinical and diagnostic parameters are summarized in Table 1. The prevalence of CDI was based on a multicenter prospective study conducted in Japan [15], where patients with at least three diarrheal bowel movements in the preceding 24 h were enrolled and tested for CDI. Hence, the prevalence of CDI represents the proportion of patients with clinically significant diarrhea who test positive for CDI. Estimates for antimicrobial treatment response for CDI were derived from clinical trials and population-based studies. The probability of recurrent healthcare onset CDI was obtained from a study based on a hospital claims database in Japan, where the recurrence of CDI increased with each subsequent CDI [24]. Excess mortality from incident and recurrent CDI were modelled with the same rates, based on a matched cohort study using a hospital claims database in Japan [26]. Based on the published literature, we modelled mortality for false negative CDI as twice that of baseline CDI mortality [27], and mortality associated with treatment failure as 3.9 times that of baseline CDI mortality [20]. The sensitivity and specificity of NAAT, GDH, and toxin antigen were derived from a single laboratory analysis of fecal specimens from patients suspected with CDI in Japanese hospitals [28]. To calculate the positive predictive value and negative predictive value of NAAT alone, and combination of GDH/toxin followed by NAAT, we used the Bayes rules of conditional probabilities [23,29].

### 2.3. Economic Data Inputs

All cost inputs shown in Table 1 were adjusted to 2022 JPY values using a web-based cost converter developed by the Campbell and Cochrane Economics Methods Group and the Evidence for Policy and Practice Information and Coordinating Centre [37]. All costs were converted to United States dollars (USD) using the same tool, where the conversions were based on the purchasing power parity (1 USD = 93.16 JPY) [37]. All cost inputs were taken from the Japanese government payer’s perspective: drug costs and diagnostic tests costs were obtained from the Japan reimbursement price list [34,35], and the remaining cost inputs were obtained from studies conducted in Japan. The cost of managing CDI was based on a study that measured the hospital-onset CDI-attributable inpatient expenditures using national level insurance claims data in Japan between 2010 to 2016 [4]. The cost of managing recurrent CDI was calculated using a multiplier of 1.91, which was obtained from a study that measured the total hospitalization costs of CDI in Japan [21]. The drug cost of intravenous metronidazole and vancomycin were added to CDI management costs for CDI treatment failure. We estimated the cost of a missed CDI diagnosis (false negative CDI) as 1.57 times the cost of CDI, by adding 7 days of hospital stay and its associated costs [8], to the baseline of 12.17 days derived from Fukuda et al. [4]. Lastly, the cost of laparoscopic colectomy from the Japanese Diagnosis Procedure Combination database was used for the cost of colectomy [36]. All calculated economic parameters are shown in Table 2.

### 2.4. Sensitivity Analyses

Deterministic one-way sensitivity analysis was performed on all clinical, diagnostic, and cost inputs to assess the independent impact of input uncertainties and overall robustness of the base case analysis. To understand the cost impact as well as the effectiveness of the new national guideline with the NAAT alone pathway, we performed sensitivity analysis on both the total budget impact and the cost per true positive CDI diagnosed. The input parameters in the base case analysis were varied by their corresponding 95% confidence intervals (CIs), or by ±25% when the 95% CIs were not available in the published literature. The range of values used for each data input are detailed in Table 1 and Table 2.

## 3. Results

### 3.1. Base Case Analysis

For 100,000 hospitalized adult patients with symptoms suspected with CDI, the NAAT alone pathway costed an additional JPY 225,886,360 (USD 2,424,714) compared to a two-step algorithm with GDH/toxin followed by NAAT. However, the NAAT alone pathway resulted in 1749 more patients accurately diagnosed with CDI and 91 fewer deaths. The cost per CDI diagnosed was JPY 26,146 (USD 281) less for the NAAT alone pathway compared to the two-step algorithm (Table 3). The incremental cost per CDI-associated death avoided was JPY 2,488,533 (USD 26,712) for the NAAT alone pathway compared to the two-step algorithm.

### 3.2. Sensitivity Analyses

The full list of results from the one-way sensitivity analysis is available in Appendix A. The total budget impact was most sensitive to changes in GDH sensitivity, followed by NAAT specificity, and the multiplier for the cost of managing missed CDI diagnosis. Cost savings of JPY 316,592,014 (USD 3,398,369) was achieved at the lower estimate of GDH sensitivity of 81.8%, and JPY 14,029,200 (USD 150,593) at a higher estimate of 1.97 for the multiplier for the cost of managing false negative CDI.

The results of the one-way sensitivity analyses for the cost per true positive CDI diagnosed are shown in Figure 3. The cost per true positive CDI diagnosed was most sensitive to the sensitivity of GDH, where the NAAT alone pathway would have a cost saving of JPY 119,827 (USD 1286) per true positive CDI patient, when GDH sensitivity was lower at 81.8%. In contrast, when GDH sensitivity was 97.9%, the NAAT alone pathway would cost an additional JPY 13,356 (USD 143) per true positive CDI patient. The model findings were also sensitive to changes in NAAT sensitivity. With a higher NAAT sensitivity of 100%, cost savings of JPY 32,809 (USD 352) per CDI accurately diagnosed was achieved with the NAAT alone pathway, while it would cost additional JPY 7572 (USD 81) per CDI accurately diagnosed when NAAT sensitivity was lower than the base case estimate at 89.9%. Changes in all other data parameters resulted in a lower cost, and hence cost saving, for each CDI accurately diagnosed using the NAAT alone pathway.

Threshold analysis showed that the NAAT alone pathway resulted in cost savings (JPY 36; USD 0.39) per true positive CDI diagnosed when GDH sensitivity was below 96.0%. This is because the NAAT alone pathway could accurately capture 805 more true positive patients. Similarly, when the sensitivity of NAAT was above 91.6%, the NAAT alone pathway would lead to cost saving (JPY 150; USD 1.61), because it could accurately capture 753 more patients.

## 4. Discussion

This is first study to evaluate the economic impact of comparing different CDI diagnostics pathways in a high disease burden population in Asia. In view of the newest addition of the NAAT pathway in January 2023 for CDI diagnosis by the Japanese Society of Chemotherapy guideline (Figure 1) [22], we conducted this budgetary impact analysis for adopting a one-step NAAT alone pathway versus a two-step diagnostic algorithm from the Japanese government payer’s perspective. In this analysis, we used mainly Japanese real world data on healthcare costs and disease epidemiology, as well as CDI attributable costs for all economic inputs, thus providing more precise and practical estimates in the budget impact analysis. The findings of this study support the adoption of the NAAT alone pathway as recommended by the latest CDI guidelines from the economic value perspective.

Our results showed that although the NAAT alone pathway was more costly compared to the two-step algorithm, it resulted in a greater number of CDI diagnosed accurately and fewer CDI-related deaths. In addition, based on our sensitivity analysis, the cost per true positive CDI diagnosed was consistently lower with the NAAT alone pathway, compared to the two-step algorithm. The adoption of NAAT for CDI diagnosis can subsequently facilitate more effective infection control measures, which has shown to reduce institutional CDI infection rates in the United States [38,39]. It is important to note that the costs generated by every true positive CDI are most likely greater than what has been calculated in the current model. This is because for every CDI infection, besides longer length of hospital stay, there is also the need for environmental decontamination, rigorous hygiene in patient care, and in some cases, cohort isolation and ward closure [40]. Therefore, the actual cost saving per true positive CDI detected by the NAAT alone pathway should be greater.

Our results are consistent with several economic evaluations that have been published on different CDI diagnostic pathways in the United States. Schroeder et al., using a CDI prevalence of 10%, reported that NAAT alone was the preferred test strategy, and was cost-effective when the cost of treating missed CDI was high (>USD 6900) [8]. Since transmission of CDI and isolation costs of CDI were both modelled [8], a missed CDI has a greater economic and clinical consequence, such that the NAAT alone pathway may remain the preferred test strategy in a setting with a lower CDI prevalence. Another cost-effectiveness analysis concluded that the NAAT alone pathway was the most cost-effective strategy for CDI diagnosis, at an incremental cost of USD 55,547 per QALY gained versus a GDH/NAAT pathway [23]. NAAT alone remained cost-effective compared with a two-step GDH/NAAT pathway over a wide range of CDI prevalence modelled (8.2–19.1%) [23].

Multiple factors affect the test performance and accuracy of CDI diagnostics, including institutional testing practices, prevalence of CDI, and asymptomatic *Clostridioides difficile* colonization [41,42]. A recent meta-analysis by the American Society for Microbiology has shown that the overall sensitivity for the detection of CDI decreased from 95% (NAAT alone) to 89% when additional tests such as GDH/toxin were added prior to NAAT [12]. Operational factors such as laboratory capacities, turnaround time for sample processing, the type of sample received, and the type of patient population tested should be considered for the adoption of NAAT technology [43]. Where a rapid, highly sensitive test is preferred, especially to minimize false negative diagnosis of CDI and deaths, NAAT alone should be prioritized to provide timely treatment and avoid CDI-associated deaths. Concerns were raised on using NAAT alone for CDI diagnosis, where patients with a NAAT+ test result may represent CDI colonization instead of an active infection [44]. However, CDI colonization is an indicator of potential transmission [45]; therefore, a NAAT+ test result is still valuable for informing the hospital that the CDI hygiene protocol should be triggered. In addition, in Japan where the prevalence of CDI exceeds 20% among symptomatic hospitalized patients [15], overdiagnosis is unlikely to occur. As previously reported in a meta-analysis by Deshpande et al., the positive predictive value for NAAT is more than 93% when CDI prevalence exceeds 20% [46].

The current budget impact analysis has a few limitations. Firstly, the cure and treatment rates for CDI were obtained from studies conducted in the United States as Japan-specific data were not available. However, sensitivity analyses revealed that these data inputs did not affect the study findings significantly. Secondly, mild-moderate and severe CDI were not modeled separately. We expect that this would have little effect on the budget impact. This is because we estimated cure rates for overall CDI using an equal proportion of metronidazole and vancomycin treatment, which was similar to the drug treatment practice in Japan [47]. Lastly, as mentioned above, the cost of missed CDI diagnosis was likely underestimated as only the costs associated with additional length of hospital stay were analyzed [8], while costs related to CDI transmission and isolation measures for CDI were not considered. We expect that the NAAT alone pathway may result in greater cost savings when these costs are considered, especially since more CDI diagnoses may be missed using a two-step algorithm. This is supported by our sensitivity analysis, where a higher treatment cost for missed CDI diagnosis resulted in overall cost savings.

## 5. Conclusions

In Japan, a significant burden of CDI exists. Timely and accurate diagnosis of CDI is key to reduce CDI-related morbidity and mortality. This study suggests that a one-step NAAT alone diagnosis pathway can result in a more accurate CDI diagnosis, lower CDI-related deaths, and lower cost per CDI diagnosed, despite having a higher cost compared to a two-step diagnosis algorithm with GDH/toxin followed by NAAT. These results are robust based on extensive sensitivity analysis. Threshold analysis showed that the NAAT alone pathway resulted in cost savings per true positive CDI diagnosed when GDH sensitivity was below 96.0%. Findings from our budget impact analysis can guide the adoption of a NAAT alone pathway for CDI diagnosis in Japan.

## Figures and Tables

**Figure 1 diagnostics-13-01463-f001:**
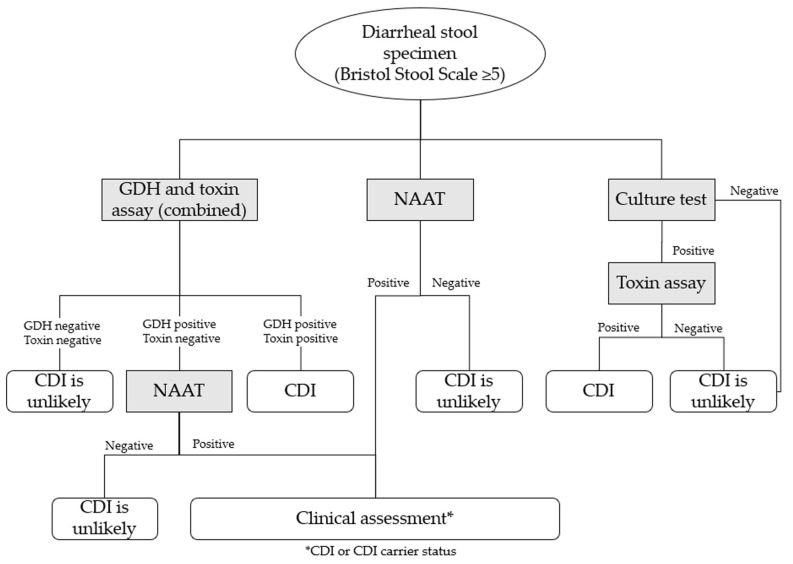
Revised *Clostridioides difficile* infection diagnostic pathway in Japan. CDI: *Clostridioides difficile* infection; GDH: Glutamate Dehydrogenase; NAAT: Nucleic Acid Amplification Test.

**Figure 2 diagnostics-13-01463-f002:**
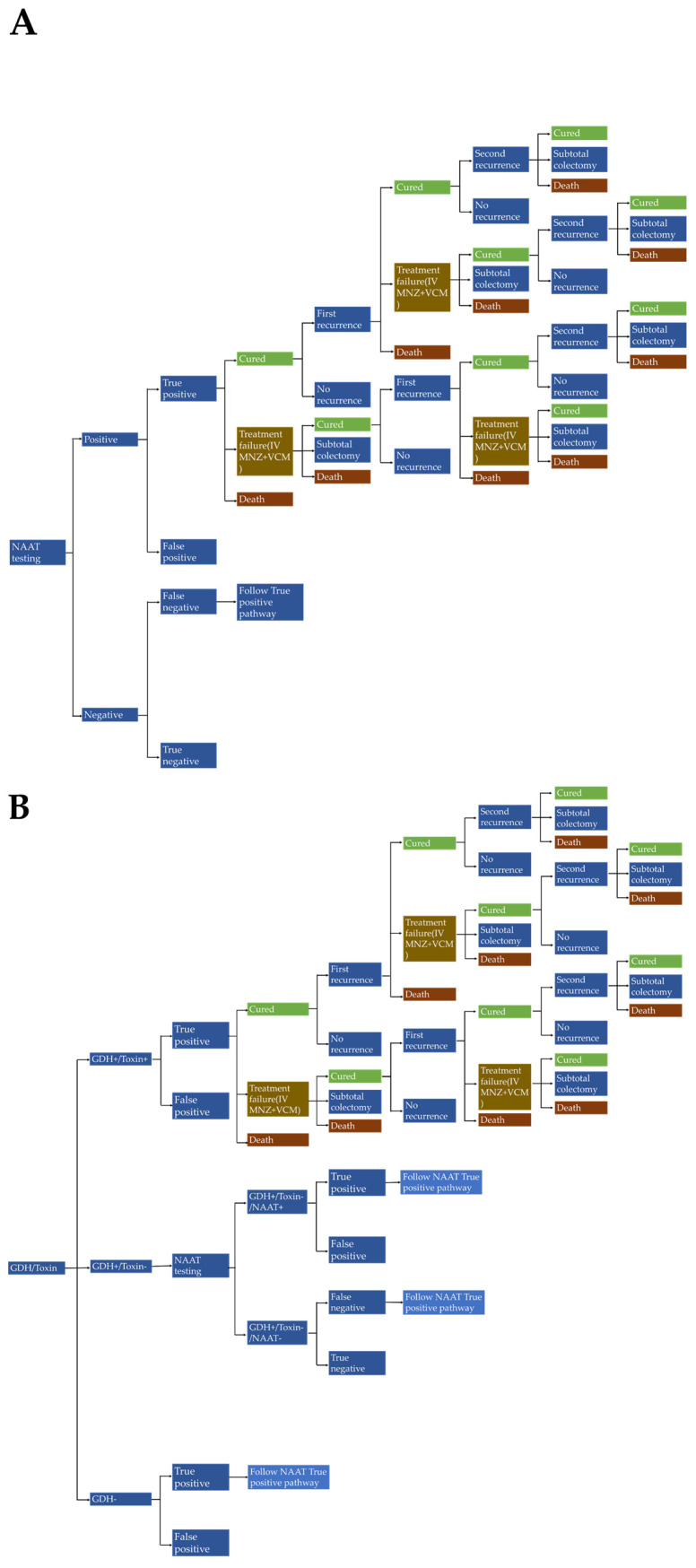
Decision tree of CDI diagnostic strategies. (**A**) Decision tree of one-step NAAT alone pathway; (**B**) Decision tree of two-step algorithm with GDH/toxin followed by NAAT. Patients with positive tests, or false negative tests would be treated. Following treatment, patients could be cured from CDI, unresponsive to treatment, or dead. Patients who failed response to the initial treatment received the next line treatment. Patients who were cured from CDI may develop another CDI, where recurrence of CDI was assumed to occur at least 4 weeks after the previous infection. Up to two recurrences were modelled in all patients, including patients who had treatment failure. CDI: *Clostridioides difficile* infection; GDH: Glutamate Dehydrogenase; IV MNZ + VCM: Intravenous Metronidazole and Vancomycin; NAAT: Nucleic Acid Amplification Test.

**Figure 3 diagnostics-13-01463-f003:**
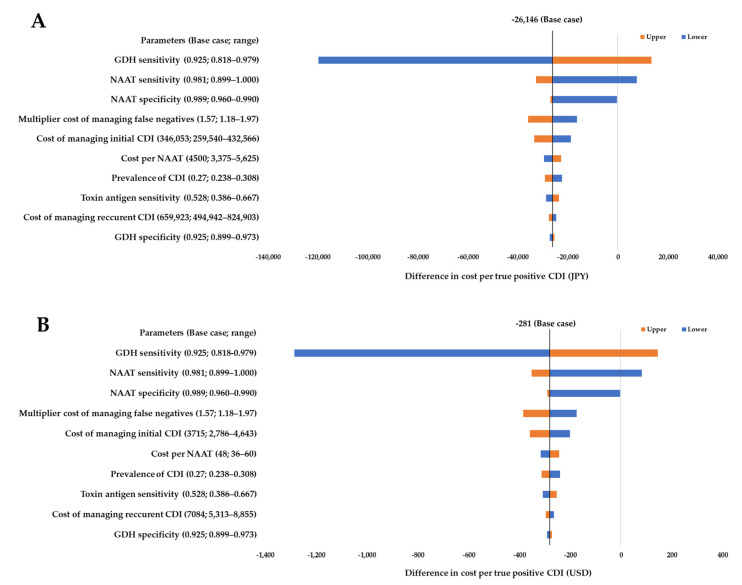
One-way sensitivity analysis for difference in cost per CDI diagnosed in the NAAT alone pathway compared to the two-step algorithm in (**A**) JPY; (**B**) USD. Data parameters that are the most sensitive to changes in the base case estimate are listed in descending order. A value below 0 would represent cost savings when the NAAT alone pathway is used compared to the two-step algorithm. CDI: *Clostridioides difficile* infection; GDH: glutamate dehydrogenase; JPY: Japanese yen; NAAT: Nucleic Acid Amplification Test; USD: United States dollar.

**Table 1 diagnostics-13-01463-t001:** Clinical, diagnostic, and economic parameters of the budget impact analysis.

Clinical Parameters	Base Case (Proportion)	Range (95%CI)	Reference
Prevalence of CDI ^1^	0.272	0.238–0.308	[15]
CDI-associated mortality	0.069	0.059–0.079	[26]
Cure from CDI treatment	0.788	0.746–0.827	[30]
Cure from recurrence (vancomycin taper and pulsed)	0.690	0.619–0.761	[31]
Cure from intravenous metronidazole and vancomycin due to treatment failure	0.841	0.733–0.949	[32]
Probability of first recurrence of CDI	0.126	0.120–0.132	[24]
Probability of second recurrence of CDI	0.227	0.205–0.251	[24]
CDI-associated mortality due to undertreatment	0.138	0.118–0.158	[27]
Increase in mortality due to treatment failure	3.90	1.40–10.7	[20]
Increase in mortality due to recurrent CDI	1.00	0.75–1.25	[26]
CDI-associated mortality due to recurrent CDI	0.0690	0.0590–0.0790	[26]
Probability of colectomy from treatment failure	0.0240	0.00500–0.0685	[33]
**Diagnostic Parameters**	**Base Case** **(Proportion)**	**Range (95%CI)**	**Reference**
NAAT sensitivity	0.981	0.899–1.00	[28]
NAAT specificity	0.989	0.960–0.990	[28]
GDH sensitivity	0.925	0.818–0.979	[28]
GDH specificity	0.944	0.899–0.973	[28]
Toxin antigen sensitivity	0.528	0.386–0.667	[28]
Toxin antigen specificity	1.00	0.979–1.00	[28]
**Economic Parameters in JPY (USD)**	**Base Case**	**Range (95%CI)**	**Reference**
Cost per NAAT test ^2^	4500 (48)	3375–5625 (36–60)	[34]
Cost per GDH/toxin test ^2^	800 (9)	600–1000 (6–11)	[34]
Total cost of managing of initial CDI	346,053 (3715)	259,540–432,566 (2786–4643)	[4]
Multiplier for cost of managing recurrent CDI	1.91	-	[21]
Additional days of hospital stay in case of false negative CDI	7	-	[8]
Cost per day due to CDI	28,435 (305)	-	[4]
Cost of drugs for treatment failure (intravenous metronidazole and vancomycin)	74,634 (801)	-	[35]
Cost of laparoscopic colectomy	1,705,371 (18,306)	1,699,709–1,711,033 (18,245–18,367)	[36]

^1^ Prevalence of CDI is defined as the proportion of patients with clinically significant diarrhea who test positive for CDI ^2^ Parameter varied by +/−25%. CDI: *Clostridioides difficile* infection; GDH: glutamate dehydrogenase; JPY: Japanese yen; NAAT: Nucleic Acid Amplification Test; USD: United States dollar.

**Table 2 diagnostics-13-01463-t002:** Calculated economic parameters of the budget impact analysis.

Economic Parameters in JPY (USD)	Base Case	Range	Remarks
Total cost of managing recurrent CDI ^1^	659,923 (7084)	494,942–824,903 (5313–8855)	Calculated from total cost of managing initial CDI and multiplier for cost of managing recurrent CDI
Multiplier for cost of managing false negatives ^1^	1.57	1.18–1.97	Calculated based on 7 additional days of hospitalization

^1^ Parameter varied by +/−25%. CDI: *Clostridioides difficile* infection; JPY: Japanese yen; USD: United States dollar.

**Table 3 diagnostics-13-01463-t003:** Cost and clinical outcomes of 100,000 patients suspected with CDI from the two different diagnostic strategies.

	Test Outcome	Cost in JPY (USD)	Number of Deaths	Total Number of True Positive Patients Diagnosed	Cost per CDI Diagnosed in JPY (USD)	Incremental Cost per Death Avoided in JPY (USD)
NAAT alone	Positive	12,731,457,123 (136,662,271)	3096	26,683	501,951 (5388)	2,488,533 (26,712)
	Negative	662,189,183 (7,108,085)	87			
	Sub Total	13,393,646,307 (143,770,355)	3183			
GDH/toxin followed byNAAT	Positive	11,610,628,467 (124,631,048)	2893	24,934	528,097 (5669)	-
	Negative	1,557,131,480 (16,714,593)	380			
	Sub Total	13,167,759,947 (141,345,641)	3274			
Difference		225,886,360 (2,424,714)	−91	1749	−26,146 (−281)	-

CDI: *Clostridioides difficile* infection; GDH: glutamate dehydrogenase; NAAT: Nucleic Acid Amplification Test.

## Data Availability

Not applicable.

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
