# Peer review of "Budget Impact Analysis of Adopting a One-Step Nucleic Acid Amplification Testing (NAAT) Alone Diagnostic Pathway for Clostridioides difficile in Japan Compared to a Two-Step Algorithm with Glutamate Dehydrogenase/Toxin Followed by NAAT"

_diagnostics, 2023, doi:10.3390/diagnostics13081463_

Round 1

Reviewer 1 Report

What is the definition of ‘true positive CDI’ in this manuscript? What is the meaning of ‘accurately diagnosed’ CDI? Please define these two concepts. These concepts should be clear before requesting of review.

Figure 1 is an incomplete pathway. Please fill in the contents.

Figure 2 has a low resolution so I can't see the contents.

Please define the prevalence of CDI in table 1. The prevalence of CDI in the reference 14 represents the CDI among CSD.

Reviewer 2 Report

1. A useful addition would be to express the costs both in yen AND an international other value, for example dollars or Euros, since the international audience is not able to understand the magnitude of the costs (and thus the benefits)

2. Why was a single evaluation of sens and spec of the methods chosen (reference 26)? is it the only Japanese relevant study? If not, why this specifically? 

Minor corrections to be made:

1. in the introduction, add  a note on the significance of antibiotic use for the emergence of CDI

2. line 53: should be "with clinically suspected"

3.  line 57 should be "versus 85% for the GDH/toxin algorithm."

4. line 58 should be "diagnoses"

5. line 59 should be "suggests"

6. line 67 "which supports..." is of unclear meaning, needs rewriting

7. line 133, should be "literature"

Round 2

Reviewer 1 Report

I have confirmed the appropriate response to the inquiry.

Thank you for your response.